# Experimental Measurement of Deposition Chloride Ions in the Vicinity of Road Cut

**DOI:** 10.3390/ma16010088

**Published:** 2022-12-22

**Authors:** Miroslav Vacek, Vít Křivý, Kateřina Kreislová, Markéta Vlachová, Monika Kubzová

**Affiliations:** 1Department of Building Structures, Faculty of Civil Engineering, VSB—Technical University of Ostrava, L. Podeste 1875, 708 00 Ostrava, Czech Republic; 2SVUOM Ltd., U Mestanskeho Pivovaru 934-4, 170 00 Prague, Czech Republic

**Keywords:** chloride, bridge, steel, reinforce concrete, road, rust, dry plate method, wet candle method, corrosion coupon

## Abstract

Chloride ions are nowadays the main cause of the degradation of steel and reinforce concrete construction in the vicinity of the road. On the other hand, chloride ions, usually in the form of de-icing salts or brine, are very important for safe winter traffic on the roads. This creates a slightly schizophrenic situation where it is necessary to ensure safe traffic in the winter period and at the same time to affect the service life of the structure as little as possible. The effect of the roadway on chloride deposition is a long-studied, but still imprecisely understood, part of the effect of chloride ions on structures in the vicinity of the roadway. This paper discusses the experimental measurement of chloride deposition in the vicinity of the I/11 road in the Czech Republic by dry plate method, wet candle method and corrosion coupons. Statistical analysis of correlation and regression is performed on the results of measurements by wet candle and horizontal dry plate methods. The methods are interdependent. A detailed analysis of the surface and chemical properties of the corrosion products is performed on the corrosion coupons. Using the corrosion loss, the environmental category C2 is determined. Observation of the microclimate in the vicinity of the roads gives to engineers a basis for the correct design of structures around the roads. The conclusions of the experimental measurements are intended to help engineers to design a structure that is safe, serviceable and sufficiently resistant to chloride ions within its service life

## 1. Introduction

For the design of steel structures, it is important to know the expected corrosion losses with respect to the reduction in the cross-section due to corrosion. Due to the effect of moisture on the surface of steel structures, a thin layer of electrolyte is formed. At the water–steel interface, a chemical reaction of water and steel occurs and thus degradation of the structure. The aggressiveness of the electrolyte is increased if sulfates or chlorides are present in the electrolyte [1]. However, due to maintenance, chloride ions enter the air in the form of aerosol, which deposits to the structures. The water droplet, together with the chloride ion, forms a local corrosion cell that can initiate pitting corrosion [2,3,4,5,6].

For designers of concrete structures, it is important to know the magnitude of chloride ion deposition. Chlorides in concrete affect the microstructure of concrete and the formation of crystals. This chemical process creates stresses around the pores of the concrete and when the limit values for the concrete used are exceeded, the concrete surface is disturbed [7,8,9,10,11,12].

However, correct design in terms of the expected degradation in the area of the structure is not the only aspect for the service life of the structure. Equally important is the consistent design of the details of the structure so that, as far as possible, they retain as little water as possible or, in the worst case, water containing chloride ions. Such detail can be a very weak point in the structure and can have fatal consequences for the overall durability and safe operation of the structure as a whole.

No less important is the effect of chloride ions on nature. Vegetation and animals living around roads are directly affected by chloride deposition. Through the transport of water through the subsoil, chloride enters the groundwater, which is the primary source of drinking water for all animals [13,14].

Historically, there has been a significant influence of sulfates on the degradation of steel structures in the Czech Republic. Due to pressure on ecology, desulfurisation units have been developed to move this influence on a marginal level [15]. Nowadays, a much more significant influence is attributed to chlorides, especially around roadways [16]. Today’s consumer society makes extensive use of transport by private cars. This is linked to the maintenance of the roads, for their good passability and to reduce the risk of accidents in the winter months [17].

In this paper, readers are presented with the results of chloride deposition and corrosion loss measurements on corrosion coupons for the year 2021 in the vicinity of the I/11 road near the village of Hrabyně-Josefovice in the Czech Republic. The aim of this article is to present the magnitude of chloride ion deposition to the readers. At the same time, the authors are looking for the dependence between the different methods of measuring chloride ion deposition, so that a suitable measurement method can be used for a particular case and the measured value can be converted to a measurement by another method if necessary. The authors also look at the effect of distance and temperature on the deposited chloride. The dependence between these variables is sought by means of a non-linear regression analysis. The paper also focuses on corrosion coupons, their analysis and the classification of the corrosion environment according to the measured corrosion losses. The magnitude of corrosion loss is compared with a standard assumption that predicts corrosion loss as a function of annual average temperature, annual average humidity, average SO_2_ and NaCl concentration.

## 2. Locality

The locality for experimental measurement was selected in vicinity road I/11 near the Hrabyně-Josefovice, the Czech Republic. Test racks are located close to the dirt road in the distance from 5 m to 180 m from the guide strip road I/11.

Two racks are located at the bridge abutment near the bridge bearings closer to the lane towards Opava, marked 0 on Figure 1. These are both equipped with a tool for a chloride deposition measurement by horizontal dry plate method only. Five racks are equipped with a stand and are located with increasing distance from the I/11 road in close proximity to the dirt road towards the village of Josefovice. The nearest of the measuring stands is at a distance of 7 m from the guide strip of the road and the furthest one is at a distance of 180 m. Distance of stands from the guide strip of the road are specified at Table 1. All these stands are equipped with tools for a chloride deposition measurement by horizontal dry plate, a vertical dry and a wet candle methods according to ISO 9225 [18]. Three of stand (on Figure 1 marked 1, 3 and 5) are equipped by horizontal and vertical corrosion coupons.

The slope of the cut of the road is approximately 30° and the depth of the cut is approximately 7 to 8 m. The slope of the cut of the road is without shrubs or trees, only grass. The slope of the road cut can be seen in Figure 2.

This measurement focuses on the wider vicinity of the road above the crown of the road notch and therefore the wider microclimate up to a distance of approximately 180 m from the guide strip of the road in the road cut.

## 3. Materials and Methods

Test methods and sample analysis were performed according to ISO 9225 [18]. A typical test stand, which can be seen in Figure 3, contained a horizontal and vertical dry plate method, wet candle method and horizontal and vertical corrosion coupons. The standard wet candle method test method was applied unchanged. In view of the experience based on the research work of Kubzová M. [19], the dry plate method test method was modified against the standard by using 100% polyester fabric from EWAC [20] instead of gauze. The dry plate method was specified in ISO 9225 [18] only as a vertical plate. In the experimental measurement by the authors a horizontal dry plate method was also used. Horizontal and vertical corrosion coupons are from steel Corten A.

The amount of chloride deposited on dry plates and wet candle methods was measured using the HACH Pocket Colorimeter™ II and cuvette tests HACH LCK311. This test kit uses ferric thiocyanate to analyse the amount of chloride in solution. The preparation of the solution for the determination of the amount of deposited chloride was carried out according to the standard ISO 9225 [18].

The corrosion coupons were subjected to post-exposure photo-documentation of the corrosion coupons, analysis of the corroded surface using a Keyence VHX-5000 microscope, analysis of the surface colour using a BYK Spectro Guide Gloss S spectrophotometer, scotch tape test, scotch tape test impression percentage analysis using a Keyence VHX-5000 microscope, corrosion product thickness analysis using a PosiTector 6000 F, elemental analysis, X-ray diffraction analysis, corrosion loss analysis according to ISO 9226 [21] and pitting analysis using a Keyence VHX-5000 microscope. Furthermore, the environmental category was assigned using the ISO 9223 [22].

## 4. Results

### 4.1. Chloride Deposition

The results of measurements near the bridge 11-134G and Hrabyně-Josefovice are shown in the Table 2. The position designation is always composed of the designation before the dash and the method used after the dash. Positions A and B are shown in Figure 1 under site designation 0, B1 is under designation 1, B2 is under designation 2. etc. The vertical dry plate method (PV), horizontal dry plate method (PH) and wet candle method (C) are used for analysing chloride deposition. The Table 2 also shows the average monthly temperature, average monthly relative humidity, thickness of the new snow layer in the month and total monthly precipitation from the station Opava-Kylešovice, managed by CHMI [23].

The value of 48.01 mg/m^2^day (marked yellow in the Table 2) in month 01/21 and site B1-PV (stand 1, vertical dry plate method) are excluded from the results due to the obvious influence of external error. Such an influence on the result may arise, for example, from vandalism, which cannot be ruled out for in situ measurements (e.g., theft of samples in subsequent months). In this case, the vandal could have touched the vertical dry plate with his hands, which had previously held snow from very close to the road. This contact would have carried a noticeably greater amount of chloride than can be spread by aerosol. The influence of the marine environment can also be excluded, as the nearest sea (the Baltic Sea) is about 500 km away [24]. The value is excluded because the average monthly chloride deposition is approximately three times higher than that measured by the other two methods, considering the conversion of the measured value between the horizontal dry plate method and the wet candle method described in ISO 9225 [18].

### 4.2. Descriptive Statistic

The normal distribution of the data was fitted with the JASP software [25]. From Table 3, it can be seen that all the data have a large skewness and kurtosis. This theory is confirmed by the Shapiro–Wilk test, or its *p*-value with a very small value (<0.001). This rejects the null hypothesis (normal distribution) and accepts the alternative hypothesis (non-normal distribution). In the following, the results of the partial statistical analyses are viewed with caution due to the non-normal distribution.

### 4.3. Statistic Corelation Analysis

The first test on the data set is to determine the causality between each set of measurements. The data are non-normally distributed, so Spearman’s correlation is used for the correlation analysis. This method is also chosen because it can partially capture some non-linear dependencies. However, correlation as such is primarily a linear statistical analysis, which is the first step in analysing data sets and is followed by more complex statistical methods of data analysis [26].

The Table 4 shows causality of each relationship. The correlation level is shown in the column “Spearman’s rho” [27].

As expected, temperature is not related to the distance from the roadside. The relationship between distance and the vertical dry plate method is affected by measurements during the summer months, when the measurements often result in measurable background values. For the rest of the values, it can be seen that at the 95% significance level the values are correlated. It is also evident from the preceding that the correlation of values is between horizontal dry plate method (PH), vertical dry plate method (PV) and wet candle method (C) with no further dependence on the location of the measurement points. The Spearman’s rho value for all three tightnesses is partially affected by the extreme values of each measurement. The relationship between the horizontal and vertical dry plate method has the highest causality value.

### 4.4. Linear Regresion

#### 4.4.1. Wet Candle Method and Vertical Dry Plate Method

The dependence of the wet candle method measurement value on the vertical dry plate method measurement value, including the proposed regression line and 95% confidence interval, is shown in Figure 4a. Q-Q plot is shown in Figure 4b.

The proposed linear regression analysis model corresponds 53.6% to the value measured by the vertical dry plate method (PV) and is therefore statistically at a significant level. The Durbin–Watson analysis is close to 2.0 (exactly 1.868), this value controls for correlation between the residuals. Both values can be found in Table 5.

The analysis of variance (ANOVA, Table 6) shows that the statistical model is below the significance level of 0.05 and the results are statistically significant.

The coefficients of the linear model are given in Table 7 and the linear regression equation is (1).
(1)C=2.908+2.700∗PV
where *C*—measured value from wet candle method (mg/m^2^day), *PV*—measured value from vertical dry plate method (mg/m^2^day).

Table 8 shows the statistics of the residuals, which indicate a normal distribution of residuals with a mean close to 0. The residue value can be observed in Figure 4b.

#### 4.4.2. Wet Candle Method and Horizontal Dry Plate Method

The dependence of the wet candle method measurement value on the horizontal dry plate method measurement value, including the proposed regression line and 95% confidence interval, is shown in Figure 5a. Q-Q plot is shown in Figure 5b.

The proposed linear regression analysis model corresponds 50.7% to the value measured by the horizontal dry plate method (PH) and is therefore statistically at a significant level. The Durbin–Watson analysis is close to 2.0 (exactly 2.002), this value controls for correlation between the residuals. Both values can be found in Table 9.

The analysis of variance (ANOVA, Table 10) shows that the statistical model is below the significance level of 0.05 and the results are statistically significant.

The coefficients of the linear model are given in Table 11 and the linear regression equation is (2).
(2)C=3.505+2.078∗PH
where *C*—measured value from wet candle method (mg/m^2^day), *PH*—measured value from horizontal dry plate method (mg/m^2^day).

The Table 12 shows the statistics of the residuals, which indicate a normal distribution of residuals with a mean close to 0. The residue value can be observed in Figure 5b.

#### 4.4.3. Vertical Dry Plate Method and Horizontal Dry Plate Method

The dependence of the vertical dry plate method measurement value on the horizontal dry plate method measurement value, including the proposed regression line and 95% confidence interval, is shown in Figure 6a. Q-Q plot is shown in Figure 6b.

The proposed linear regression analysis model corresponds 80.4% to the value measured by the vertical dry plate method (PV) and is therefore statistically at a significant level. The Durbin–Watson analysis is close to 2.0 (exactly 1.941), this value controls for correlation between the residuals. Both values can be found in Table 13.

The analysis of variance (ANOVA, Table 14) shows that the statistical model is below the significance level of 0.05 and the results are statistically significant.

The coefficients of the linear model are given in Table 15 and the linear regression equation is (3).
(3)PH=0.295+1.098∗PV
where *PH*—measured value from horizontal dry plate method (mg/m^2^day), *PV*—measured value from vertical dry plate method (mg/m^2^day).

The Table 16 shows the statistics of the residuals, which indicate a normal distribution of residuals with a mean close to 0. The residue value can be observed in Figure 6b.

#### 4.4.4. Comparison of the Linear Regression Analysis with the Assumptions from ISO 9225

Three equations were established by regression analysis to describe the relationship between the vertical dry plate method, the horizontal dry plate method, and the wet candle method measurements. ISO 9225 [18] only addresses this issue for the vertical dry plate and wet candle measurement values. In ISO 9225 [18], the approximate Equation (4) is given. If the Equation (1) from the linear regression analysis is compared to the Equation (4), they have very similar trend. The constant value of 2.908 and the difference in multiplicity of 0.300 from the vertical dry plate method measurements may be due to the gauze used in the wet candle method, the fabric used in the dry plate method, the climatic conditions of the last days of the month before the test samples were changed, and many other random phenomena or random or systematic errors in the analysis of the exposed samples. Data will continue to be refined based on additional data obtained from planned long-term chloride deposition measurements in the vicinity of the I/11.
(4)$$S_{d,c}=2.4*S_{d,p}$$
where *S_d,c_*—measured value from wet candle method (mg/m^2^day), *S_d,p_*—measured value from vertical dry plate method (mg/m^2^day).

### 4.5. Non-Linear Regresion

Non-linear regression analysis is performed for the measured values using the vertical dry plate method. The average monthly temperature and the distance from the road guide strip were found to be appropriate as independent variables [28,29].

A nonlinear regression model with the least sum of squares residuals is evaluated in Equation (5).
(5)$$PV=max\{\begin{array}{c} x*e^{a+temp}+y*dist^{b}-const\\ min \end{array}\;$$
where *PV*—value of vertical dry plate method (mg/m^2^day), *a*, *b*, *x*, *y*—coefficients from non-linear regression analysis, *e*—Euler’s number, *temp*—average monthly temperature (°C), *dist*—distance from the road guide strip (m), *const*—constant and *min*—minimum value for non-linear equation.

The solution of the nonlinear regression analysis is performed using Microsoft Excel software and Solver add-in. In the case of non-linear analysis, the R^2^ is not the telling factor, so the SSR is observed in this analysis [30].

The minimum value is proposed as the average of the 04/21 to 10/21 measurements at sites B1-PV to B5-PV, all these values are quantified in the Table 17. Minimum value for non-linear equation is expressed by the Equation (6). This value can be considered as a measurable background value.
(6)$$min=\frac{\sum PV_{i}}{n}=1.14$$
where *min*—minimum value for non-linear equation, *PV_i_*—value of vertical dry plate method (mg/m^2^day), *n*—number of values.

The coefficients for the linear regression are found by minimizing the SSR using Microsoft Excel and the Solver add-in. Their values are expressed in the Table 18.

In the statistical non-linear regression analysis, the minimum and maximum value of the residual is observed in Table 19.

The residual is obtained by subtracting the model calculation value from the measured value. The residual value is important for the subsequent sum of the squared residuals (hereafter SSR), which determines the quality of the calculation model. The minimum and maximum calculation value is also observed and compared with the measured values, see Table 20.

Final equation is (7). Calculated values of deposition of chloride ions for individual months are quantified in the Table 21. A graphical representation of the measured and calculated values is shown in Figure 7.
(7)$$PV=max\{\begin{array}{c} -109.27*e^{-2.93+temp}+36.42*dist^{-0.53}+7.26\\ 1.14, \end{array}$$
where *PV*—value of vertical dry plate method (mg/m^2^day), *temp*—average monthly temperature (°C) and *dist*—distance from the road guide strip (m).

### 4.6. Corrosion Coupons

The exposure of the corrosion coupons was carried out in the period 3 November 2020–20 December 2021 and the samples were exposed in positions B1 (horizontal only, the vertical one was damaged by vandals), B3 and B5.

#### 4.6.1. Analyse the Surface of the Corrosion Coupons

A Keyence VHX-5000 microscope was used to analyse the surface of the corrosion coupons.

The surface of the horizontally placed corrosion coupons can be seen in Figure 8. The condition of the whole sheet is shown, 50×, 100× and 200× magnification. The large white spots are bird droppings, the remaining surface has a consistent patina. With increasing distance, it is possible to observe the loss of deposited salt (white dots). At 200× magnification it is possible to see flaking pieces of corroded surface in the first two samples (B1-H—7 m, B3-H—45 m). The size of the flaking pieces is about 1 mm.

The surface of the vertically placed corrosion coupons can be seen in Figure 9. The condition of the whole sheet is shown, 50×, 100× and 200× magnification. The surface of the corrosion coupons has a uniform patina.

#### 4.6.2. Thickness of the Corrosion Products

The Table 22 and Figure 10 shows the thickness of the corrosion products. The measurements were made using a PosiTector 6000 F. The thickness of the corrosion products was measured at 30 locations on the corrosion coupon.

#### 4.6.3. Colorimetric Analysis

The exposed corrosion coupons were analysed for colour by the BYK Spectro Guide Gloss S. The colour values in the L, a, b coordinate system are given in the Table 23. To illustrate the colour scheme, a diagram of the L, a, b colour spectrum of the CIELab system is shown in Figure 11 [31]. Sometimes the coordinates L*, a*, b* are used, which are the same as L, a, b.

The pre-exposure colour values are not analysed because the surface of the exposed corrosion coupons before exposure is metallic glossy without corrosion product layers. The measured data will be compared with other corrosion coupons or with results from real steel structures in future research.

#### 4.6.4. Scotch Tape Test

The scotch tape test was performed on an area of 70 × 70 mm [32]. The results are shown in Figure 12. On the corrosion coupon exposed in a horizontal position at a distance of 7 m from the road guide strip, the particles are the coarsest. Their number increases with increasing distance but they become finer. A similar trend can be observed for the vertical orientation of the corrosion coupons.

The percentage of marks from the Scotch tape test is shown in Figure 13. To cover as much area as possible, a magnification of 20× is chosen. The analysis of the area is performed using a Keyence VHX-5000 microscope. The reported percentage of marks values are mean value from the four observed scotch tape test surfaces for each corrosion coupon.

#### 4.6.5. Average Corrosion Increment and Average Corrosion Loss

The results of the average corrosion increment and loss measurements after 1 year of exposure are shown in the Table 24 and Figure 14. The mass increment is determined by gravimetric methods in accordance with ISO 8407 [33]. Corrosion increment is the average thickness of corrosion products measured by the magnetic-induction method. The measurement results are shown in the Table 22.

Using Table 2 in ISO 9223 [22] and the results of the corrosion loss analysis, the environmental category C2 is determined [34]. The interval values for the corrosion rate r_corr_ are 10–200 g/m^2^year and 1.5–25 µm/year. In all exposure cases the measured values are between the limit values for environmental category C2.

Although there are differences between the locations and positions of the corrosion coupons, the determined values lead to the same corrosivity category C2 according to ISO 9223. The limit for the corrosivity category C2 is from 1.3 µm/year to 25 µm/year [22].

#### 4.6.6. Normative Corrosivity

Normative corrosivity estimation based on calculated first-year corrosion loses from ISO 9223 [22] is the Equations (8) and (9).
(8)$$r_{corr}=1.77*P_{d}^{0.52}*exp\left(0.020*RH+f_{St}\right)+0.102*S_{d}^{0.62}*exp\left(0.033*RH+0.040*T\right)$$
(9)$$f_{St}=0.150*\left(T-10\right)\;when\;T\leq 10{}^{\circ}C,\;otherwise-0.054*\left(T-10\right)$$
where *r_corr_*—first year corrosion rate of metal (µm/year), *T*—annual average monthly temperature (°C), *RH*—annual average relative humidity (%), *P_d_*—annual average SO_2_ deposition (mg/m^2^day), *S_d_*—annual average Cl^−^ deposition (mg/m^2^day).

The average SO_2_ concentration is taken from the final report ZU Ostrava for the year 2021 [35]. Values at the two closest measuring stations to the corrosion coupons are shown in the Table 25. Distance from measuring stations to location of corrosion coupons is in Figure 15.

The value for the calculation was determined by engineering estimation considering the limits of SO_2_ deposition in the urban environment (5–100 µg/m^3^) according to ISO 9223 [22] and the location of the site, which is rural environment. The selected SO_2_ concentration value is therefore 3.00 µg/m^3^ (limit of 2–15 µg/m^3^ for rural environment).

The relationship between the volumetric SO_2_ concentration and the SO_2_ deposition rate is given in the Equation (10).
(10)$$P_{d}=0.8*P_{c}$$
where *P_d_*—annual average SO_2_ deposition (mg/m^2^day), *P_c_*—annual average SO_2_ volumetric method (µg/m^3^).

The T and RH values are calculated by a weighted arithmetic mean of the Table 2 values. The results are shown in the Table 26.

The S_d_ value is calculated as a weighted arithmetic mean of the wet candle chloride deposition measurements (Table 2) according to ISO 9223 [22] for each location. The results are shown in the Table 27.

The resulting r_corr_ values for the individual corrosion coupon location are given in the Table 28. When compared with the results in the Table 24, the values are very close with horizontally placed corrosion coupons. The value from the experimental measurement of the horizontal position of the corrosion coupon at site B3 is larger than the calculated value, but it is not a significant difference.

#### 4.6.7. Pits

The size, diameter, and number of pits on the corrosion coupon are shown in the Table 29, Figure 16 and Figure 17. The average, minimum and maximum pitting depth and diameter values are from five observations of each corrosion coupon with the Keyence VHX-5000 microscope. The number of pits is collected over an area of 1792.68 × 1344.51 µm in five observations of each corrosion coupon and then this is converted to the number of pits per 1 mm^2^.

An observation of the surface after chemical cleaning procedures for removal of corrosion products of the horizontal corrosion coupons is shown in Figure 18. The image of the whole corrosion coupon B5-H (Figure 18c) shows the colour artefact on the surface. The diameter, depth and number of pits are measured outside this area. Also shown is a 200× magnification of the sample and an axonometric view of the surface after chemical cleaning procedure for removal of corrosion products. In the three aforementioned views at 200× magnification, the size and depth, diameter and number of pits can be observed. Visually, the most severe pitting of the surface is evident at the corrosion coupon located closest to the road guide strip (B1-H). As the distance increases, a less severe pitting of the surface can be observed. According to the results in the Table 29, the pits at site B1-H are deeper, have a larger diameter and are the largest in number of the horizontal corrosion coupons analysed. The horizontal corrosion coupons from sites B3-H and B5-H show very similar average, minimum and maximum values of depth, diameter and number of pits.

An observation of the surface after chemical cleaning procedures for removal of corrosion products of the vertical corrosion coupons is shown in Figure 19. A 200× magnification of the sample and an axonometric view of the surface after chemical cleaning procedures for removal of corrosion products is shown. The size and depth, diameter and number of pits on the vertical corrosion coupons can be observed. Visually, the two surfaces look very similar. This trend is confirmed by the results from the Table 29. The vertical corrosion coupons from sites B3-V and B5-V show very similar average, minimum and maximum values of depth, diameter and number of pits.

#### 4.6.8. Elemental Analysis

The results of the elemental analysis are summarized in the Table 30. For this paper, the weight percentages of chlorine (Cl) and sulfur (S) are important. In the table, it can be observed that the weight fraction of chlorine on the horizontal corrosion coupons decreases with increasing distance from the road guide strip. It is worth noting the values of the mass percentage of chlorine on the vertical corrosion coupons, which is higher than that of the horizontal corrosion coupons. It can also be observed that the percentage by mass of sulfur increases with increasing distance from the road guide strip. This may be due to the decreasing distance to the village of Hrabyně-Josefovice, where some inhabitants may use fossil fuels (mainly coal) for heating, thus releasing sulfur dioxide (SO_2_) into the air [36,37].

#### 4.6.9. X-ray Diffraction Analysis

The results of the X-ray diffraction analysis are shown in the Table 31. The main components of weathering steel corrosion products include goethite, akaganeite, lepidocrocite, magnetite and maghemite [38,39].

Lepidocrocite is the major phase in all corrosion coupons. This phase is the least stable phase and is present at the beginning of the weathering steel exposure. All corrosion coupons have goethite as the second phase, which is the most stable phase of corrosion products. Goethite protects against penetration by the main causes of corrosion, i.e., oxygen and water, and the corrosion stimulants, which are chlorides. [40] This contributes to the protective properties of the patina. A minority phase is akaganeite, which is found in weathering steel surfaces that are exposed in environments where chlorides are present.

Protective ability indices (PAI_α_, PAI_β_) indicate the quality of protection of corrosion products [41,42,43]. When the PAI_α_ value is greater than 1.0, the corrosion coating can be considered as having a corrosion rate of less than 10 µm/year. If the PAI_α_ is less than 1.0, the PAI_β_ index must be evaluated. For PAI_β_ index greater than 0.5, a protective layer is formed with increasing corrosion rate. A PAI_β_ index less than 0.5 indicates that a corrosion layer without protective function is formed. [38,39] The function of the PAI_β_ index is affected by the phase of akaganeite formed in the presence of chloride ions. The results of the X-ray diffraction analysis show that the PAI_α_ index is less than 1.0 in all cases evaluated, so the focus should be on the PAI_β_ index which is less than 0.5. Thus, a corrosion layer without protective function is formed on the corrosion coupons.

## 5. Discussion

The first half of the paper summarizes the measurements of chloride deposition in the vicinity of the road I/11 near the Hrabyně-Josefovice. This paper presents chloride ion deposition data measured from November 2020 to December 2021 from wet candle and dry plate method measurements. The measured dates are accompanied by data on climatic conditions from the meteorological station Opava-Kylešovice, which is under the administration of CHMI.

The wet candle and dry plate method measurements show a similar trend to the authors’ previous measurements [19,44]. Chloride deposition occurs mainly in the winter period, whereas in the summer periods the measured deposition is at the background values. It can be assumed that depending on the average temperatures in the winter period, the chloride deposition will vary due to the amount of thawing salt applied.

The thickness of the corrosion products, visual analysis of the surface and results of other analyses show similar results to the authors’ earlier measurements [19,44]. Therefore, it can be concluded that under similar weather conditions and similar chloride deposition (i.e., similar location), the result observed on the corrosion coupons will correlate with the presented measurement.

### 5.1. Statistic Analysis

From the measured values, the correlation between the wet candle method, the horizontal dry plate method and the vertical dry plate method is demonstrated. The magnitude of the Spearmen’s rho correlation between the wet candle and dry plate methods is around 0.3. The vertical and horizontal dry plate methods have a higher Spearmen’s rho correlation coefficient, specifically 0.838.

The interdependence is also quantified using linear regression analysis. The relationship between the vertical dry plate method and the wet candle method is given in Equation (1). The regression analysis of the relationship between these two methods shows a similar trend to the equation in ISO 9225. The relationship between the wet candle method and the horizontal dry plate method is given in Equation (2). This relationship shows a similar trend to Equation (1) for the wet candle method and the vertical dry plate method. Therefore, given the high correlation coefficient, it can be expected that the horizontal and vertical dry plate methods will have very similar measured chloride ion deposition values. This assertion is proven, and the resulting Equation (3) shows their correlation, whereby multiplying the measured value of the vertical dry plate method by 1.098 and adding the value of 0.295, the value of the horizontal dry plate method is calculated. This result makes the assumption that horizontal surfaces of structures (e.g., the top surface of the bottom flange of steel beams of a steel-concrete bridge) will have approximately 10% higher measured chloride deposition than vertical surfaces (e.g., the webs of steel beams of a steel–concrete bridge). This observation leads the authors to conduct further research, which will, among other things, also address this direction.

Furthermore, a non-linear regression analysis is performed on the measured values to demonstrate the dependence of the measured value of the vertical dry plate method on the average monthly temperature and the distance of the measuring station from the guide strip of the road. The resulting Equation (7) shows results very similar to the measured values in situ. The resulting non-linear relationship shows that the amount of chloride emitted increases with decreasing temperature along an exponential curve. At the same time, as the distance decreases, the value of deposited chloride increases following a power function with a negative exponent. However, this relationship can only be applied over the crown of the road. The authors will continue to investigate the wider surroundings of the road and validate the measured data. However, they will also focus on the closer surroundings of the roadway, where even higher chloride deposition values are expected.

### 5.2. Corrosion Coupons

The corrosion coupons are subjected to visual analysis, analysis with a Keyence VHX-5000 microscope. The visual analysis provides a view of the entire coupon and magnification images. None of the coupons show visual abnormalities. The size of the flaking corrosion products can be observed in the magnified images. However, the size of these corrosion products is small. This is mainly due to the short exposure time of the corrosion coupons. However, it is one of the important visual information that can be a comparison for longer exposed corrosion coupons.

For the exact measurement of the colour of corrosion products, the colorimetric analysis with a BYK Spectro Guide Gloss S colorimeter is used. Colorimetric values prior to exposure of the corrosion coupons are not available because the coupons are metallic shiny at the beginning of the exposure. The results of the analysis can be used by other corrosion engineers for comparison with their own measurements using the colorimetric method. The authors will continue to use the colorimetric method and compare the values from the one-year exposure with other corrosion coupons that will be exposed in subsequent years. In further research, the authors will also focus on the immediate vicinity of the roadway where the corrosion coupons will be exposed. At the same time, corrosion coupons that have been exposed for a longer period of time are also installed. A subsequent comparison with these colorimetric measurement values will also be carried out on the named corrosion coupons.

Corrosion layer thickness is measured with a PosiTector 6000 F. The horizontally oriented corrosion coupon located 7 m from the road guide strip has the highest thickness of corrosion products, exactly average corrosion layer thickness is 73.7 µm. The horizontal coupons at 45 m and 180 m have very similar corrosion product thicknesses (55.9 µm and 56.9). For the vertical orientation of the corrosion coupons, a greater thickness of corrosion products can be observed for the coupon at 45 m from the guide strip of the road, exactly 61.2 µm. Vertical corrosion coupon B5-V have average corrosion layer thickness 51.4 µm. The results show that the thickness of the corrosion layer decreases with increasing distance. It is also possible to observe the different behaviour of horizontal and vertical corrosion coupons, with the vertical corrosion coupon having a larger corrosion layer closer to the emitter and the horizontal corrosion coupon farther from the emitter. The results show that the thickness of the corrosion layer decreases with increasing distance. It is also possible to observe the different behaviour of horizontal and vertical corrosion coupons, where closer to the emitter the vertical corrosion coupon has a larger corrosion layer and farther from the emitter the horizontal corrosion coupon has a larger corrosion layer.

Next analysis is scotch tape test. On the corrosion coupon exposed in a horizontal position at a distance of 7 m from the road guide strip, the particles are the coarsest. Their number increases with increasing distance but they become finer. A similar trend can be observed for the vertical orientation of the corrosion coupons. Scotch tape test of horizontal corrosion coupons shows a percentage of area covered by corrosion product particles between 9% and 14%. The vertical position of the corrosion coupons shows an area covered by corrosion product particles between 8% and 10%. Thus, it is evident that the percentage of loose parts is less in the case of vertical orientation of the exposed surface.

In this paper, the corrosion loss at sites B1, B3 and B5 is determined. The corrosion loss ranges from approximately 90 to 160 g/m^2^ and 10 to 20 µm, respectively. From the analysis of the average corrosion gain and loss, the environmental category C2 was determined using Table 2 in ISO 9223. This standard loss is determined on the basis of chloride deposition measurements by the authors and SO_2_ concentration measurements by ZU Ostrava. The measured and calculated corrosion loss have similar values. This was a verification of the reality and prediction model of the standard.

The pitting analysis shows that the pits at site B1 on the horizontal corrosion coupon are the largest and deepest. At the same time, this corrosion coupon has the largest number of pits in the area. As the distance increases, the pits are smaller, less deep and their number decreases or stagnates.

The results of the elemental analysis showed that the largest proportion of chloride ions is in the corrosion products of sample B3-V. This sample has the highest proportion of akaganeite (2%, X-ray diffraction analysis), which is mainly formed by the action of chloride ions on corrosion coupons. Another corrosion coupon that has appreciable akageneite is B1-H. This corrosion coupon also has a large proportion of chloride ions in the corrosion products. The other coupons have a smaller proportion of chloride ions and the amount of akageneite is minor. The results of the X-ray diffraction analysis show that the PAI_α_ index is less than 1.0 in all cases evaluated, so the focus should be on the PAI_β_ index which is less than 0.5. Thus, a corrosion layer without protective function is formed on the corrosion coupons.

## 6. Conclusions

In the case of steel structures, the corrosivity predicts the corrosion loss over the life cycle of the structure. For reinforced concrete structures, chloride ions cause faster carbonation of concrete, which reduces the service life of reinforced concrete structures. This paper summarizes the results of chloride deposition measurements and analyses on corrosion coupons after one year of exposure in the vicinity of the I/11 road near the village of Hrabyně-Josefovice in the Czech Republic. The results of the measurements can help corrosion engineers and designers of structures in the vicinity of roadways to properly design structures for their service life design.

Three measurements of annual chloride ion exposure have already been carried out at the Hrabyně locality. In each year the climatic conditions were different, especially the amount of snow and the number of days with an average daily temperature below 4 °C, when de-icing salts and brine are applied due to icy roads. Even with repeated measurements, the difference in the measured amount of chloride deposition and corrosion loss on the corrosion coupons is not statistically significant. The results can be considered representative for similar localities.

An important conclusion of this work is that the different methods of measuring chloride deposition are correlated. However, correlation alone does not describe their relationship. This relationship is expressed by the results of a linear regression analysis. The authors will make follow-up measurements and refine the correlation between the different measurement methods.

An interesting finding is the non-linear relationship between the average monthly temperature, the distance from the road guide strip and the magnitude of chloride deposition measured by the vertical dry plate method. This relationship is applied to a measured sample of data and the authors will further investigate the effect of each input variable on the accuracy of the results of this relationship.

This paper summarizes the results of chloride deposition measurements and analyses on corrosion coupons after one year of exposure in the vicinity of the I/11 road near the village of Hrabyně-Josefovice in the Czech Republic. The authors’ team will continue to measure chloride deposition around roads. On the basis of the published results, an extension of the measurement sites in the vicinity of Ostrava has been carried out to cover the closer vicinity of roads in the cut and fill. Validation of the already known results will be carried out on the basis of further data obtained.

## Figures and Tables

**Figure 1 materials-16-00088-f001:**
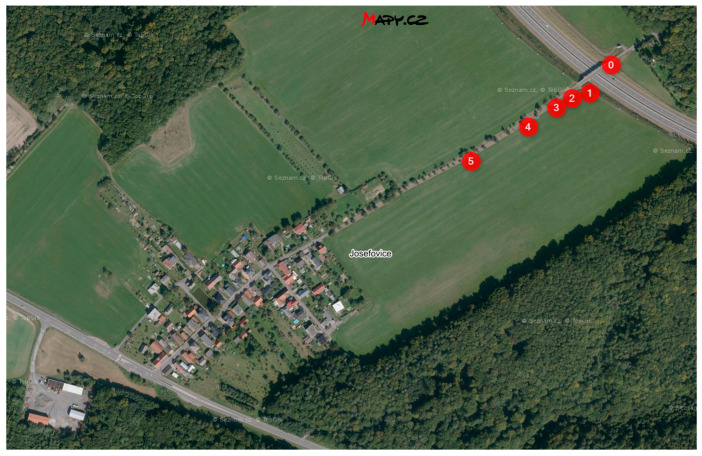
Locality near Hrabyně-Josefovice and bridge 11-134G (source: mapy.cz).

**Figure 2 materials-16-00088-f002:**
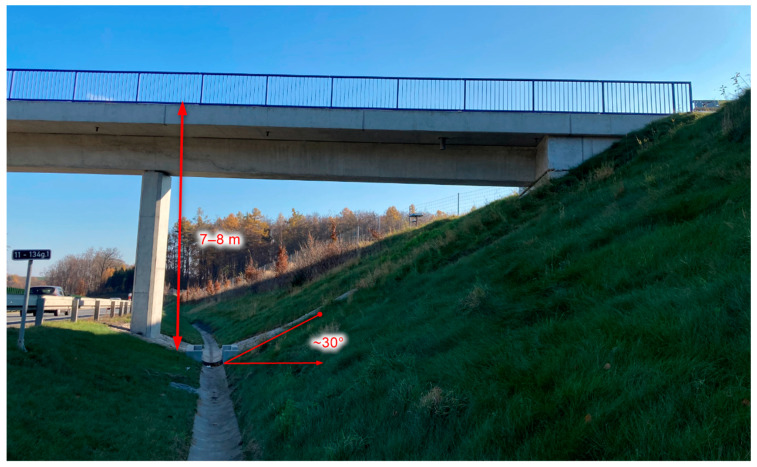
Locality near Hrabyně–Josefovice and bridge 11-134G.

**Figure 3 materials-16-00088-f003:**
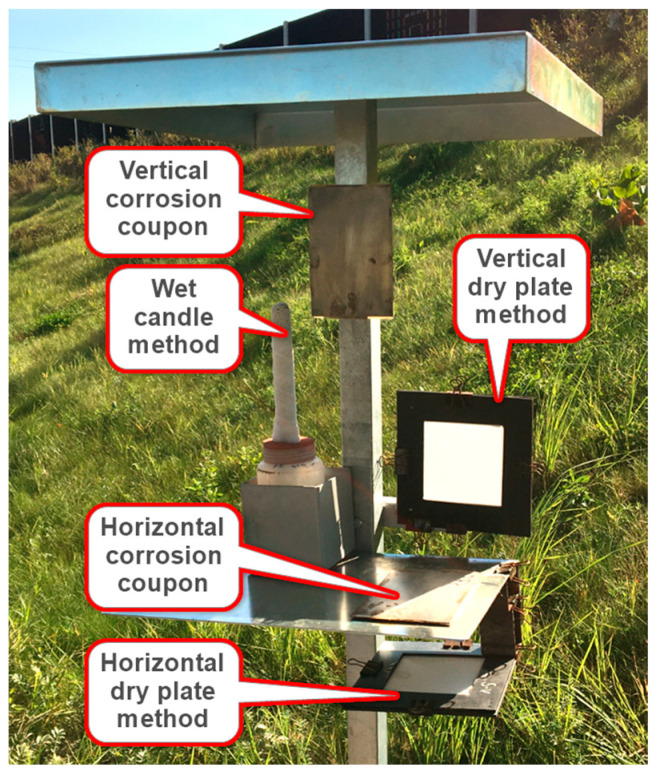
Typical stand for experimental measurement of deposition chloride ions.

**Figure 4 materials-16-00088-f004:**
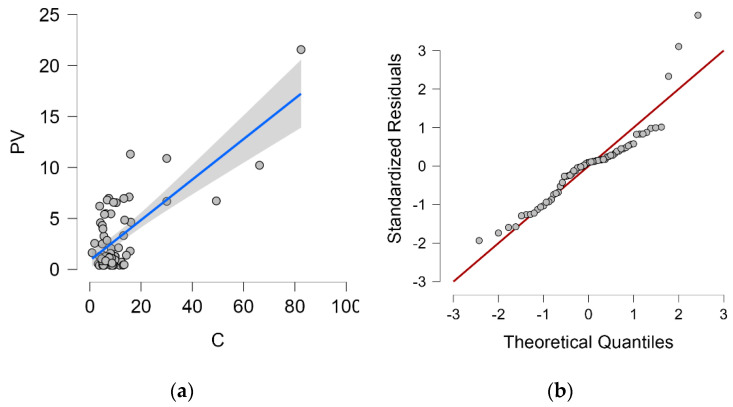
Vertical dry plate method and wet candle method. (**a**) Dependence of vertical dry plate method values and 95% confidence interval (grey area) on wet candle method values; (**b**) Q-Q plot.

**Figure 5 materials-16-00088-f005:**
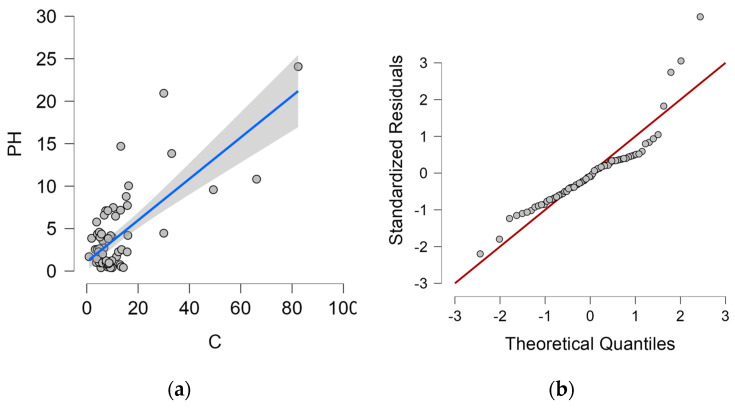
Wet candle method and horizontal dry plate method. (**a**) Dependence of horizontal dry plate method values and 95% confidence interval (grey area) on wet candle method values; (**b**) Q-Q plot.

**Figure 6 materials-16-00088-f006:**
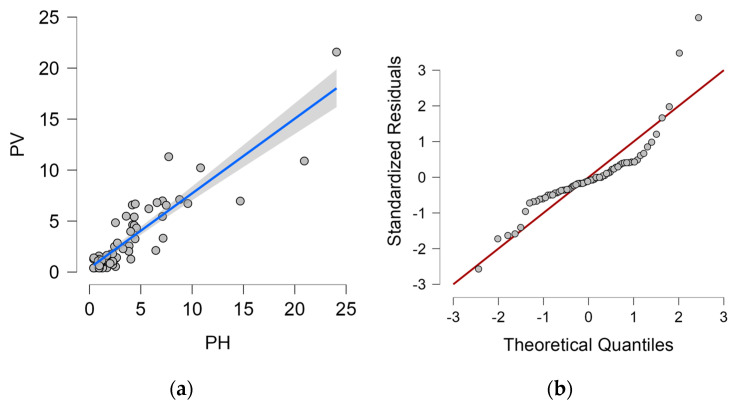
Horizontal dry plate method and vertical dry plate method. (**a**) Dependence of horizontal dry plate method values on and 95% confidence interval (grey area) vertical dry plate method values; (**b**) Q-Q plot.

**Figure 7 materials-16-00088-f007:**
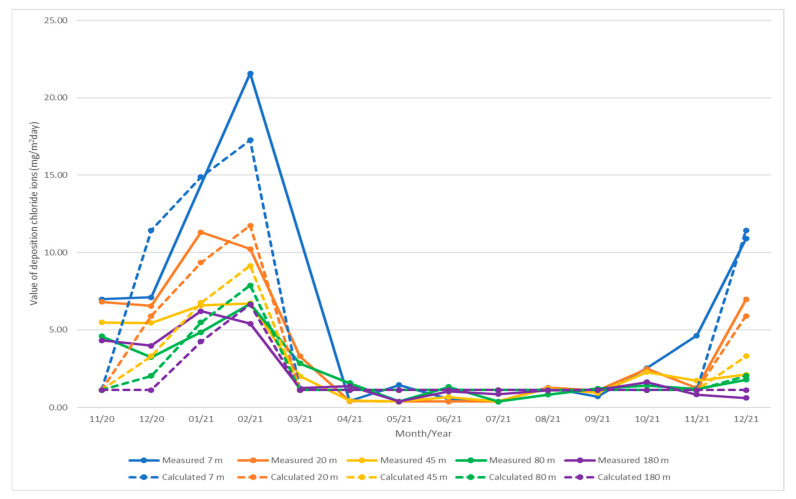
Graphical representation of the measured and calculated values of deposition of chloride ions (mg/m^2^day).

**Figure 8 materials-16-00088-f008:**
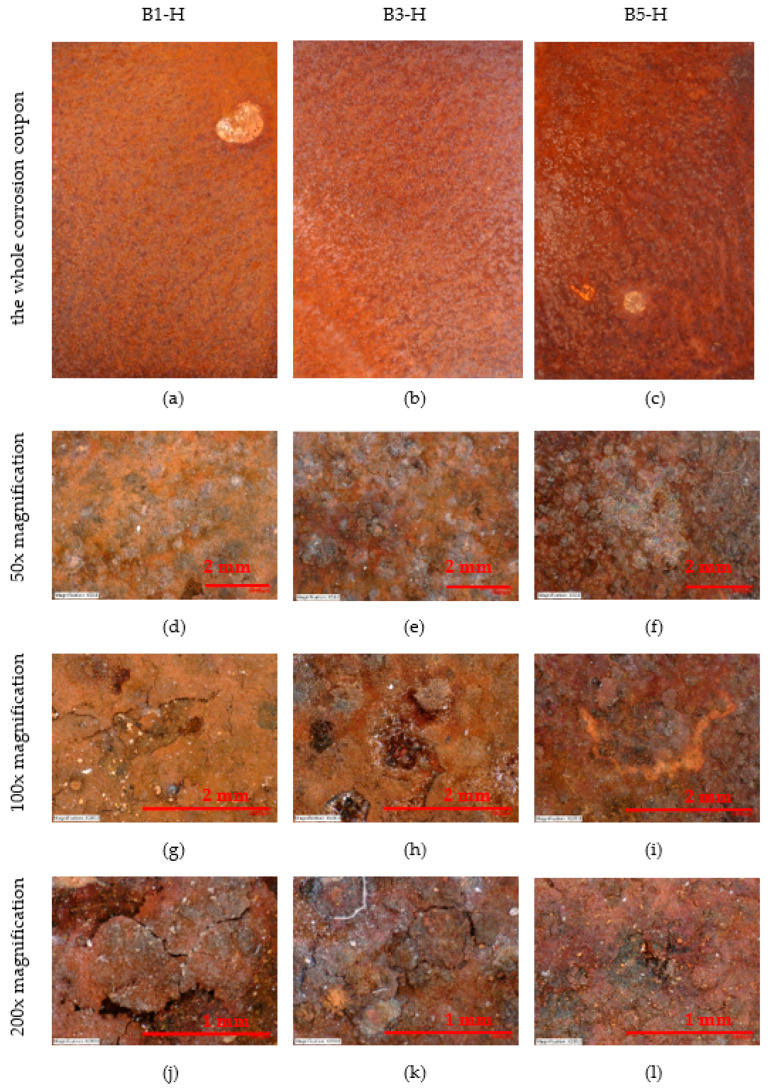
Appearance of horizontal corrosion coupons on stands—50×, 100× and 200× magnification. (**a**–**c**) The whole corrosion coupons; (**d**–**f**) 50× magnification of corrosion coupon surface; (**g**–**i**) 100× magnification of corrosion coupon surface; (**j**–**l**) 200× magnification of corrosion coupon surface.

**Figure 9 materials-16-00088-f009:**
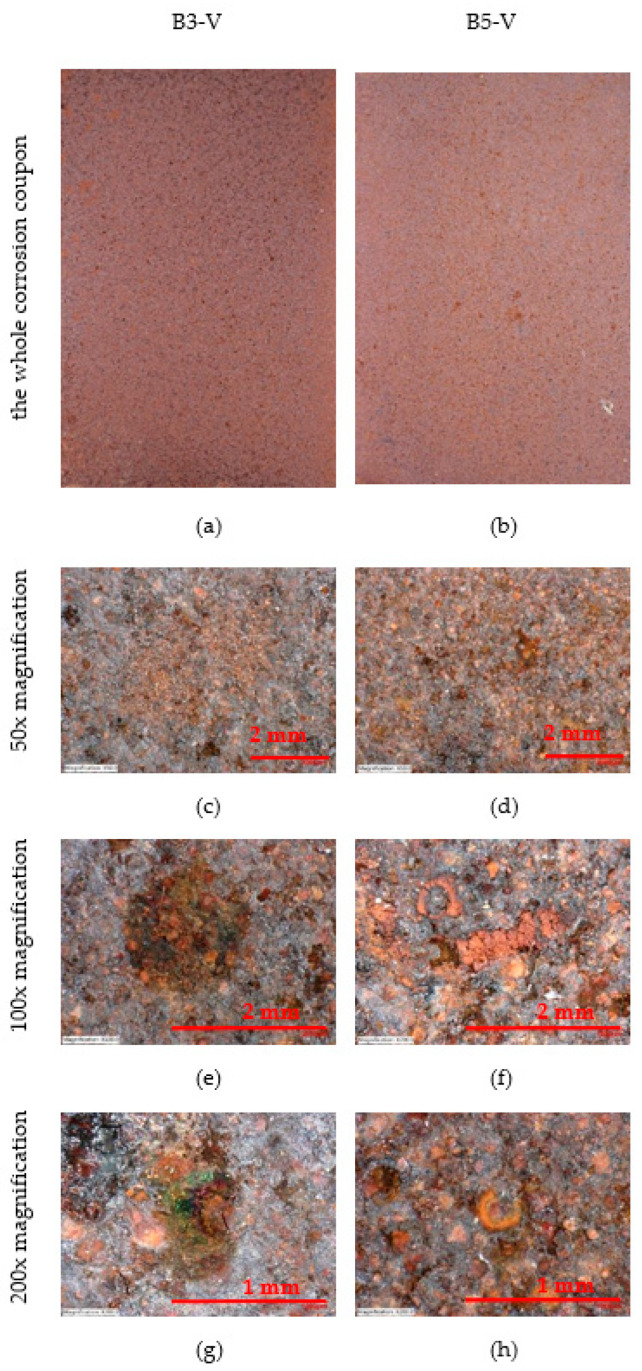
Appearance of vertical corrosion coupons on stands—50×, 100× and 200× magnification. (**a**,**b**) The whole corrosion coupons; (**c**,**d**) 50× magnification of corrosion coupon surface; (**e**,**f**) 100× magnification of corrosion coupon surface; (**g**,**h**) 200× magnification of corrosion coupon surface.

**Figure 10 materials-16-00088-f010:**
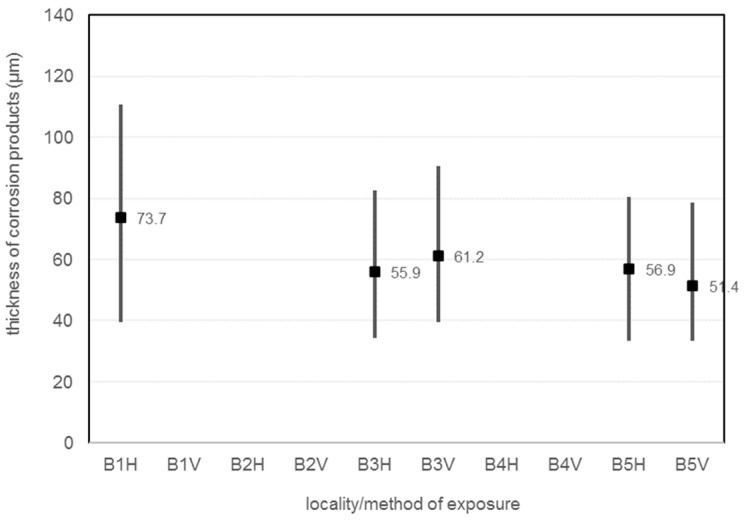
Thickness of corrosion products.

**Figure 11 materials-16-00088-f011:**
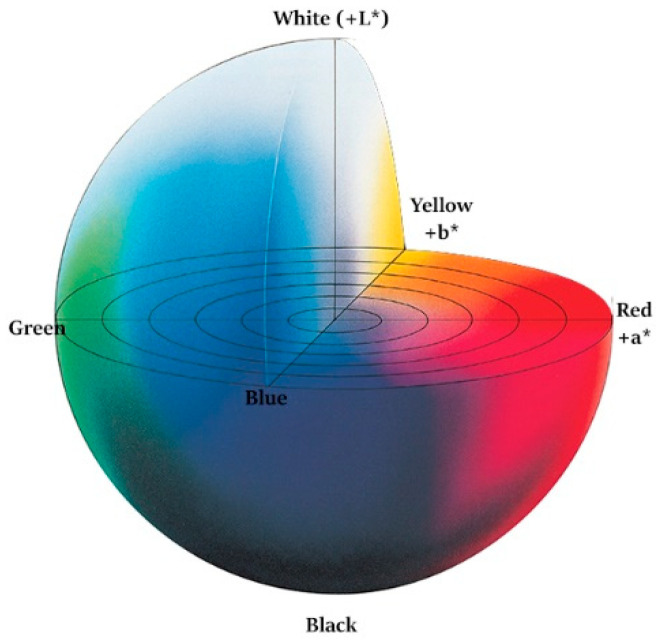
Schematic of the colour spectrum L, a, b of the CIELab system [31] (Photo courtesy of Konica Minolta, Inc. All rights reserved).

**Figure 12 materials-16-00088-f012:**
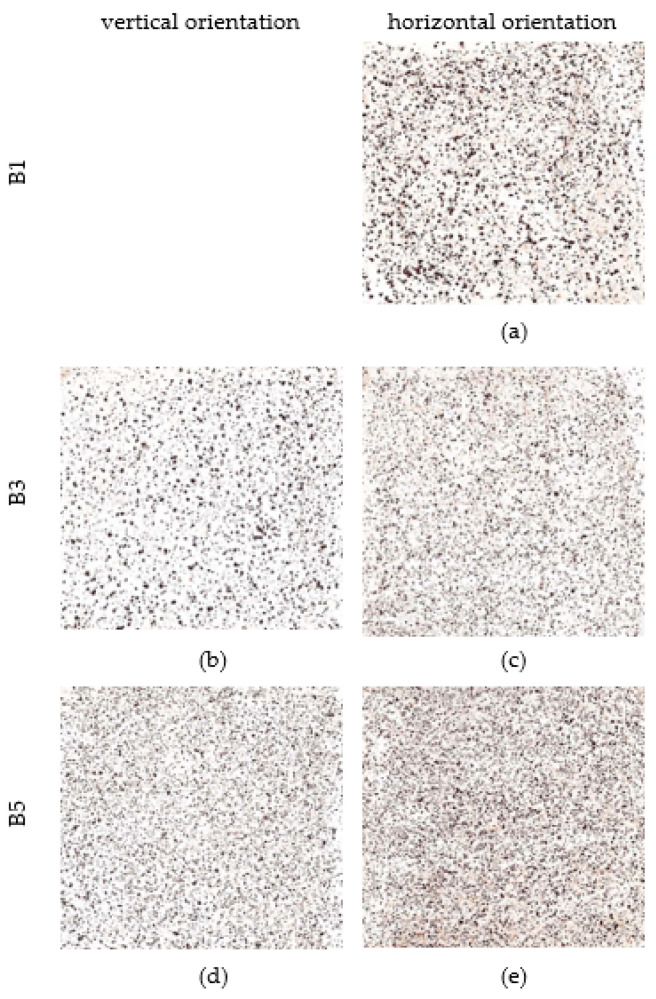
Scotch tape test (area 70 × 70 mm). (**a**,**c**,**e**) Scotch tape test—horizontal orientation of corrosion coupons from stand B1, B3 and B5 (**b**,**d**) Scotch tape test—vertical orientation of corrosion coupons from stand B3 and B5.

**Figure 13 materials-16-00088-f013:**
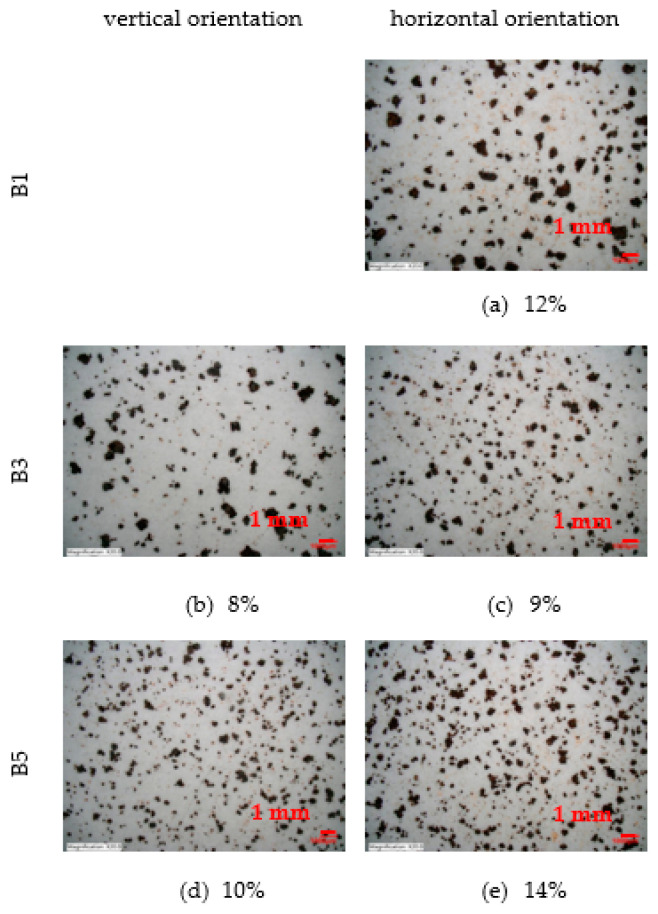
Percentage of marks from Scotch tape test (20× magnification). (**a**,**c**,**e**) Percentage of marks from scotch tape test—horizontal orientation of corrosion coupons from stand B1, B3 and B5 (**b**,**d**) Percentage of marks from scotch tape test—vertical orientation of corrosion coupons from stand B3 and B5.

**Figure 14 materials-16-00088-f014:**
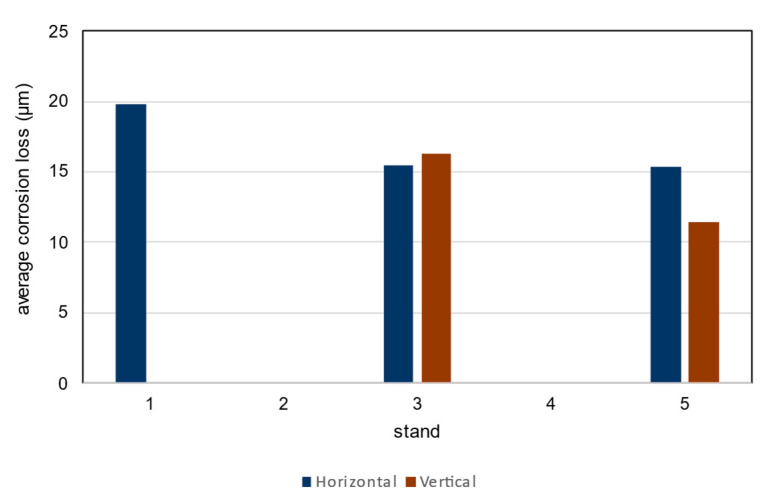
Average corrosion loss after 1 year of exposure.

**Figure 15 materials-16-00088-f015:**
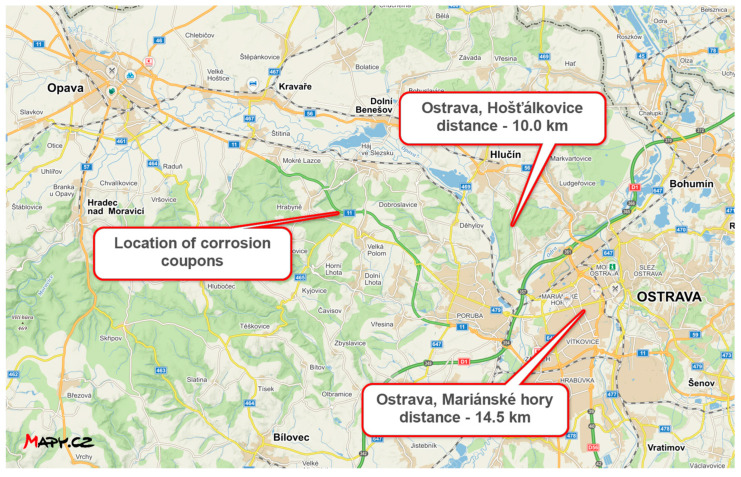
Distance between the position of the corrosion coupons and the CHMI measuring stations (source: mapy.cz).

**Figure 16 materials-16-00088-f016:**
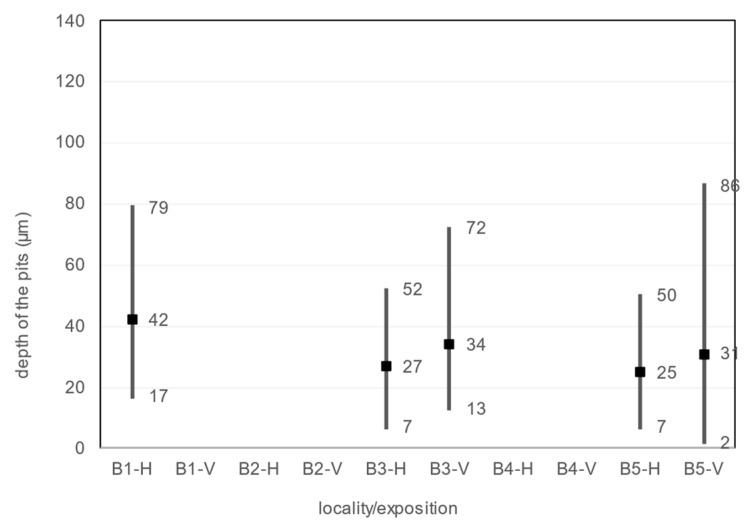
Depth of the pits.

**Figure 17 materials-16-00088-f017:**
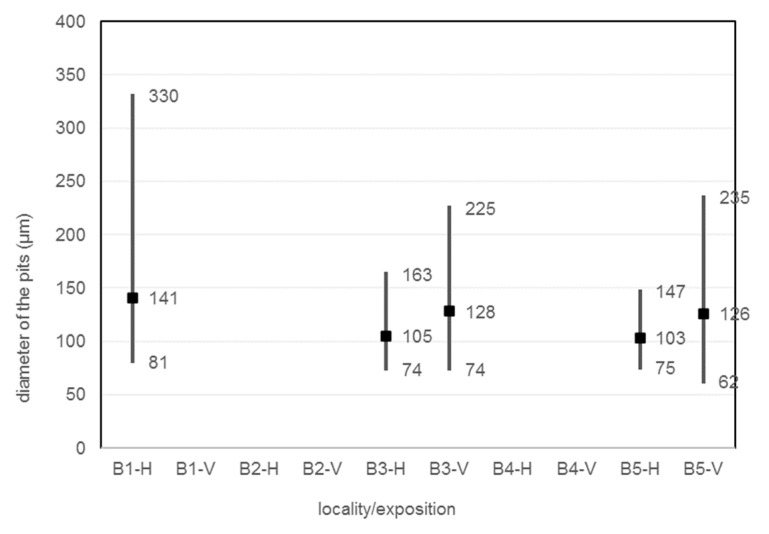
Diameter of the pits.

**Figure 18 materials-16-00088-f018:**
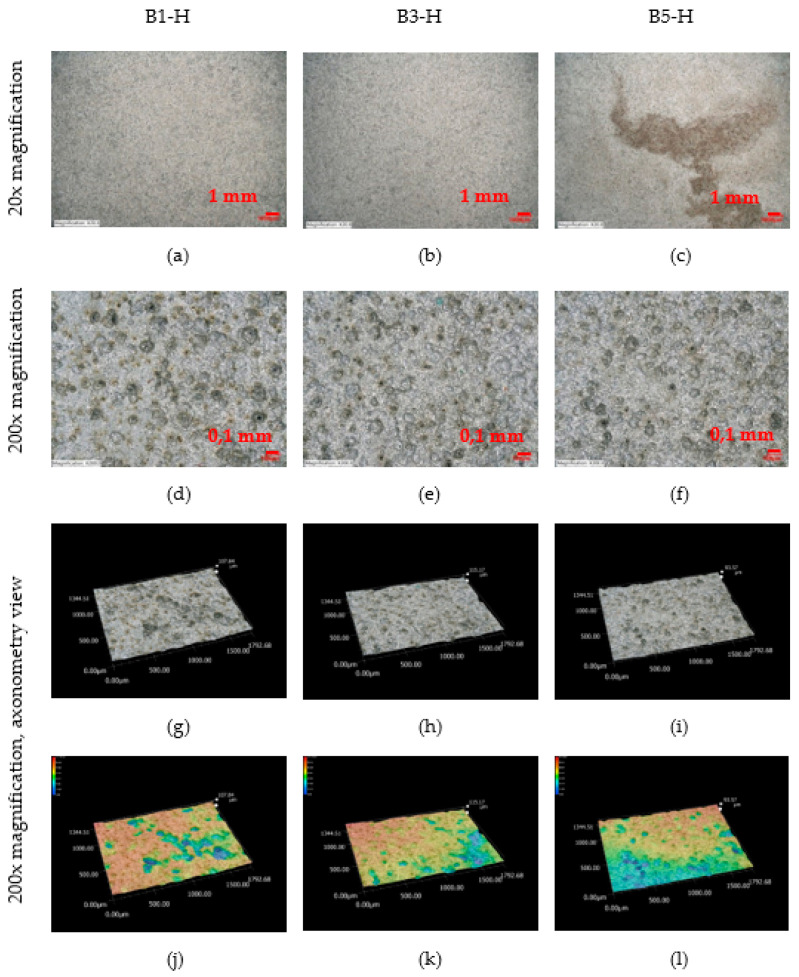
(**a–l**) Structure of the steel surfaces under the corrosion layer—horizontal exposure.

**Figure 19 materials-16-00088-f019:**
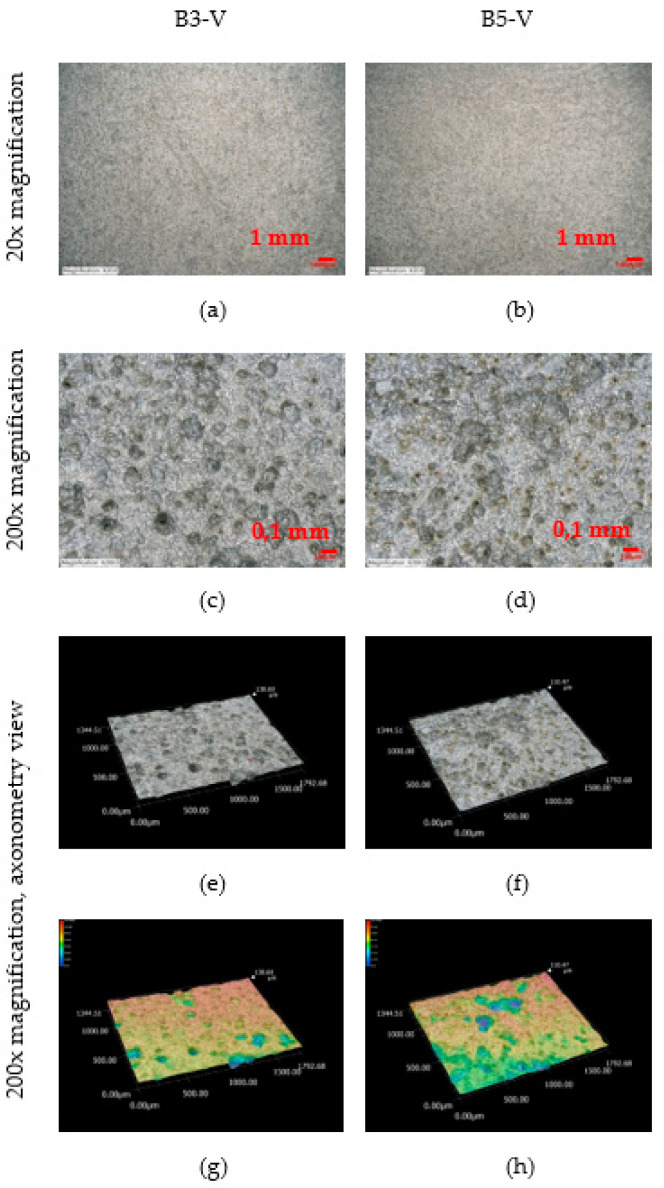
(**a–h**) Structure of the steel surfaces under the corrosion layer—vertical exposure.

**Table 1 materials-16-00088-t001:** Distance of stands from the guide strip of the road.

Stand	0	1	2	3	4	5
Distance (m)	5	7	20	45	80	180

**Table 2 materials-16-00088-t002:** Average monthly value of deposition chloride ions (mg/m^2^day).

**Temperature (°C)**		4.8	0.4	−0.1	−0.7	3.7	6.4	12.2	19.4	20.4	17.3	14.6	9.1	4.8	0.4
**Relative** **humidity (%)**		82	83	83	80	72	70	69	66	70	75	75	72	82	83
**New snow (cm)**		25	8	20	20	2	3	0	0	0	0	0	0	25	8
**Rainfall (mm)**		42.3	22.5	40.1	41.0	26.4	64.8	110.3	72.9	67.8	172.9	31.3	20.1	50.2	35.7
**Days of exposure**		30	32	31	24	30	29	31	31	33	31	31	28	31	34
**Position**	**Distance**	**11/20**	**12/20**	**1/21**	**2/21**	**3/21**	**4/21**	**5/21**	**6/21**	**7/21**	**8/21**	**9/21**	**10/21**	**11/21**	**12/21**
**A-PH**	**5 m**	3.65	7.92	29.68	25.96	4.09	1.27	2.03	Miss.	0.77	0.84	2.65	4.74	4.49	14.30
**B-PH**	**5 m**	3.48	6.85	30.21	13.02	5.15	1.49	1.60	Miss.	0.40	1.32	2.35	2.84	3.82	8.54
**B1-C**	**7 m**	7.38	15.36	33.08	82.37	16.29	11.58	7.70	3.28	5.56	9.37	12.35	1.89	16.01	30.00
**B1-PH**	**7 m**	7.11	8.78	13.84	24.07	10.04	1.70	1.05	2.54	0.40	0.74	2.27	3.86	4.20	20.93
**B1-PV**	**7 m**	6.98	7.11	48.01	21.57	Miss.	0.42	1.47	0.54	0.40	1.29	0.72	2.56	4.63	10.89
**B2-C**	**20 m**	6.78	10.31	15.82	66.18	13.14	12.97	9.10	3.70	5.50	9.91	7.35	4.87	9.82	13.26
**B2-PH**	**20 m**	6.58	7.48	7.72	10.82	7.17	0.75	1.16	1.01	0.40	0.40	1.00	2.44	4.03	14.69
**B2-PV**	**20 m**	6.81	6.55	11.31	10.22	3.32	0.42	0.40	0.40	0.40	1.28	1.10	2.50	1.27	6.96
**B3-C**	**45 m**	6.22	8.22	9.26	49.33	8.39	13.38	8.09	4.99	5.08	8.38	9.79	Miss.	Miss.	11.20
**B3-PH**	**45 m**	3.58	7.11	4.17	9.58	3.83	0.51	0.49	1.13	1.28	0.67	1.22	3.26	2.03	6.46
**B3-PV**	**45 m**	5.48	5.45	6.58	6.72	2.04	0.46	0.40	0.69	0.40	1.22	0.94	2.29	1.74	2.12
**B4-C**	**80 m**	4.15	5.48	13.58	30.00	6.72	8.06	9.25	4.10	5.32	4.77	7.31	4.96	3.93	15.73
**B4-PH**	**80 m**	4.36	4.44	2.53	4.46	2.72	0.92	0.40	2.55	0.96	1.07	0.86	2.64	1.45	2.26
**B4-PV**	**80 m**	4.58	3.26	4.84	6.68	2.85	1.58	0.40	1.33	0.40	0.84	1.23	1.42	1.19	1.80
**B5-C**	**180 m**	4.88	5.11	3.84	5.72	6.11	14.26	8.77	4.58	7.52	8.47	7.58	0.88	6.17	8.68
**B5-PH**	**180 m**	4.58	4.01	5.78	4.36	0.96	0.42	0.94	2.27	1.02	0.85	1.21	1.69	2.01	0.97
**B5-PV**	**180 m**	4.32	3.98	6.21	5.40	1.26	1.39	0.40	1.04	0.87	1.11	1.15	1.64	0.84	0.62

Note: Miss. = Missing measurement.

**Table 3 materials-16-00088-t003:** Descriptive statistic using JASP.

	Temperature	Distance	C	PH	PV
Valid	98	98	68	96	68
Missing	0	0	30	2	30
Mean	8.050	48.857	11.753	4.778	3.098
Std. Deviation	7.263	59.604	13.455	6.072	3.582
Skewness	0.429	1.405	3.649	2.608	2.588
Std. Error of Skewness	0.244	0.244	0.291	0.246	0.291
Kurtosis	−1.246	0.650	14.863	7.234	9.754
Std. Error of Kurtosis	0.483	0.483	0.574	0.488	0.574
Shapiro–Wilk	0.884	0.712	0.553	0.669	0.719
*p*-value of Shapiro–Wilk	<0.001	<0.001	<0.001	<0.001	<0.001
Minimum	−0.700	5.000	0.880	0.400	0.400
Maximum	20.400	180.000	82.370	30.210	21.570

**Table 4 materials-16-00088-t004:** Correlation analysis using JASP.

			n	Spearman’s Rho	*p*
Temperature	-	Distance	98	0	1
	-	C	68	−0.493	<0.001
	-	PH	96	−0.782	<0.001
	-	PV	68	−0.779	<0.001
Distance	-	C	68	−0.365	0.002
	-	PH	96	−0.293	0.004
	-	PV	68	−0.136	0.27
C	-	PH	68	0.273	0.024
	-	PV	66	0.313	0.011
PH	-	PV	68	0.838	<0.001

**Table 5 materials-16-00088-t005:** Vertical dry plate method and wet candle method—model.

					Durbin–Watson		
Model	R	R^2^	Adjusted R^2^	RMSE	Autocorrelation	Statistic	*p*
H₀	0	0	0	13.385	0.096	1.806	0.428
H₁	0.732	0.536	0.529	9.184	0.045	1.868	0.55

**Table 6 materials-16-00088-t006:** Vertical dry plate method and wet candle method—ANOVA.

Model		Sum of Squares	df	Mean Square	F	*p*
H₁	Regression	6246.871	1	6246.871	74.069	<0.001
	Residual	5397.7	64	84.339		
	Total	11,644.572	65			

**Table 7 materials-16-00088-t007:** Vertical dry plate method and wet candle method—coefficient of linear regression analysi.

							95% CI	
Model		Unstandardized	Standard Error	Standardized	t	*p*	Lower	Upper
H₀	(Intercept)	11.361	1.648		6.896	<0.001	8.071	14.651
H₁	(Intercept)	2.908	1.498		1.942	0.057	−0.084	5.9
	PV	2.7	0.314	0.732	8.606	<0.001	2.073	3.326

**Table 8 materials-16-00088-t008:** Vertical dry plate method and wet candle method—statistic of residuals.

	Minimum	Maximum	Mean	SD	N
Predicted Value	3.988	61.141	11.361	9.803	66
Residual	−17.622	35.681	−2.32 × 10^−16^	9.113	66
Std. Predicted Value	−0.752	5.078	4.71 × 10^−17^	1	66
Std. Residual	−2.015	4.037	0.011	1.04	66

**Table 9 materials-16-00088-t009:** Wet candle method and horizontal dry plate method—model.

					Durbin–Watson		
Model	R	R^2^	Adjusted R^2^	RMSE	Autocorrelation	Statistic	*p*
H₀	0	0	0	13.455	0.23	1.539	0.053
H₁	0.712	0.507	0.499	9.522	−0.012	2.002	0.953

**Table 10 materials-16-00088-t010:** Wet candle method and horizontal dry plate method—ANOVA.

Model		Sum of Squares	df	Mean Square	F	*p*
H₁	Regression	6146.569	1	6146.569	67.798	<0.001
	Residual	5983.574	66	90.66		
	Total	12,130.144	67			

**Table 11 materials-16-00088-t011:** Wet candle method and horizontal dry plate method—coefficient of linear regression analysis.

							95% CI	
Model		Unstandardized	Standard Error	Standardized	t	*p*	Lower	Upper
H₀	(Intercept)	11.753	1.632		7.203	<0.001	8.496	15.01
H₁	(Intercept)	3.505	1.529		2.293	0.025	0.453	6.557
	PH	2.078	0.252	0.712	8.234	<0.001	1.574	2.582

**Table 12 materials-16-00088-t012:** Wet candle method and horizontal dry plate method—statistic of residuals.

	Minimum	Maximum	Mean	SD	N
Predicted Value	4.336	53.523	11.753	9.578	68
Residual	−20.771	40.191	1.21 × 10^−16^	9.45	68
Std. Predicted Value	−0.774	4.361	8.17 × 10^−17^	1	68
Std. Residual	−2.294	4.325	0.005	1.043	68

**Table 13 materials-16-00088-t013:** Horizontal dry plate method and wet candle method—model.

					Durbin–Watson		
Model	R	R^2^	Adjusted R^2^	RMSE	Autocorrelation	Statistic	*p*
H₀	0	0	0	4.385	0.324	1.337	0.005
H₁	0.897	0.804	0.801	1.956	−0.096	2.188	0.468

**Table 14 materials-16-00088-t014:** Horizontal dry plate method and wet candle method—ANOVA.

Model		Sum of Squares	df	Mean Square	F	*p*
H₁	Regression	1035.899	1	1035.899	270.853	<0.001
	Residual	252.423	66	3.825		
	Total	1288.322	67			

**Table 15 materials-16-00088-t015:** Horizontal dry plate method and wet candle method—coefficient of linear regression analysis.

							95% CI	
Model		Unstandardized	Standard Error	Standardized	t	*p*	Lower	Upper
H₀	(Intercept)	3.696	0.532		6.95	<0.001	2.634	4.757
H₁	(Intercept)	0.295	0.315		0.937	0.352	−0.333	0.923
	PV	1.098	0.067	0.897	16.458	<0.001	0.965	1.231

**Table 16 materials-16-00088-t016:** Horizontal dry plate method and wet candle method—statistic of residuals.

	Minimum	Maximum	Mean	SD	N
Predicted Value	0.734	23.973	3.696	3.932	68
Residual	−4.99	8.681	−1.20 × 10^−16^	1.941	68
Std. Predicted Value	−0.753	5.157	−2.70 × 10^−17^	1	68
Std. Residual	−2.679	4.641	0.000916	1.019	68

**Table 17 materials-16-00088-t017:** Measured values for determining the minimum model value.

	3/21	4/21	5/21	6/21	7/21	8/21	9/21	10/21
B1-PV	Miss.	0.42	1.47	0.54	0.40	1.29	0.72	2.56
B2-PV	3.32	0.42	0.40	0.40	0.40	1.28	1.10	2.50
B3-PV	2.04	0.46	0.40	0.69	0.40	1.22	0.94	2.29
B4-PV	2.85	1.58	0.40	1.33	0.40	0.84	1.23	1.42
B5-PV	1.26	1.39	0.40	1.04	0.87	1.11	1.15	1.64

Note: Miss. = Missing measurement.

**Table 18 materials-16-00088-t018:** Variables for non-linear regression equation.

Variables	
a	−2.93
b	−0.53
x	−109.27
y	36.42
const	7.26
min	1.14
SSR	215.77

**Table 19 materials-16-00088-t019:** Minimum and maximum value of residuum.

Residuum	
min	0.00
max	5.84

**Table 20 materials-16-00088-t020:** Minimum and maximum value of measured and calculated value.

	Measured	Calculation
min	0.40	1.14
max	21.57	17.268

**Table 21 materials-16-00088-t021:** Calculated value of deposition of chloride ions (mg/m^2^day).

	11/20	12/20	01/21	02/21	03/21	04/21	05/21	06/21	07/21	08/21	09/21	10/21	11/21	12/21
**Temperature (°C)**	4.8	0.4	−0.1	−0.7	3.7	6.4	12.2	19.4	20.4	17.3	14.6	9.1	4.8	0.4
**Calculated 7 m**	14.32	14.32	14.32	14.32	14.32	14.32	14.32	14.32	14.32	14.32	14.32	14.32	14.32	14.32
**Calculated 20 m**	8.79	8.79	8.79	8.79	8.79	8.79	8.79	8.79	8.79	8.79	8.79	8.79	8.79	8.79
**Calculated 45 m**	6.19	6.19	6.19	6.19	6.19	6.19	6.19	6.19	6.19	6.19	6.19	6.19	6.19	6.19
**Calculated 80 m**	4.93	4.93	4.93	4.93	4.93	4.93	4.93	4.93	4.93	4.93	4.93	4.93	4.93	4.93
**Calculated 180 m**	3.69	3.69	3.69	3.69	3.69	3.69	3.69	3.69	3.69	3.69	3.69	3.69	3.69	3.69

**Table 22 materials-16-00088-t022:** Thickness of corrosion products.

			Thickness (µm)			
Stand/Exposition		n	Mean	Min	Max	s_x_
B1	horizontal	30	73.7	40	110	15.6
B3	horizontal	30	55.9	35	82	13.5
	vertical	30	61.2	40	90	10.3
B5	horizontal	30	56.9	34	80	12.5
	vertical	30	51.4	34	78	11.0

**Table 23 materials-16-00088-t023:** Colorimetric parameters of exposed samples.

Stand/Exposition		Coordinates CIELab		
		L	a	b
B1	horizontal	30.43	12.08	16.33
B3	horizontal	27.75	11.26	12.71
	vertical	34.04	6.02	3.75
B5	horizontal	26.10	12.75	13.55
	vertical	33.63	8.14	6.90

**Table 24 materials-16-00088-t024:** Average corrosion increment and loss after 1 year of exposure.

Stand/Exposition		Average Corrosion Gain		Average Corrosion Loss	
		(g.m^−2^)	(µm)	(g.m^−2^)	(µm)
B1	horizontal	89.95	73.7	155.78	19.8
B3	horizontal	68.93	55.9	121.92	15.5
	vertical	73.83	61.2	127.90	16.3
B5	horizontal	58.98	56.9	120.81	15.4
	vertical	48.33	51.4	89.99	11.4

**Table 25 materials-16-00088-t025:** Average concentration of SO_2_ in 2021 [35] and value for calculation r_corr_.

Station	Average Concentration SO_2_ P_c_ (µg/m^3^)	Average Concentration SO_2_ P_d_ (mg/m^2^)
Ostrava, Mariánské Hory	<11.00	<8.80
Ostrava, Hošťálkovice	<11.00	<8.80
Value for calculation	3.00	2.40

**Table 26 materials-16-00088-t026:** Weighted arithmetic means of RH and T, calculated f_St_.

RH (%)	T (°C)	f_St_
75.88	8.20	−0.27

**Table 27 materials-16-00088-t027:** Weighted arithmetic means of measuring by wet candle method for location B1, B3 and B5.

S_d,B1-C_ (mg/m^2^day)	S_d,B3-C_ (mg/m^2^day)	S_d,B5-C_ (mg/m^2^day)
17.15	9.57	6.65

**Table 28 materials-16-00088-t028:** Normative corrosivity estimation based on calculated first-year corrosion loses.

r_corr,B1_ (µm/Year)	r_corr,B3_ (µm/Year)	r_corr,B5_ (µm/Year)
19.80	16.74	15.32

**Table 29 materials-16-00088-t029:** Depth of the pits, diameter of the pits, average number of pits.

Stand/Exposition		Depth of the Pits (µm)			Diameter of the Pits (µm)			Average Number of Pits (Pits/mm^2^)
		average	min.	max.	average	min.	max.	
B1	horizontal	42	17	79	141	81	330	23.5
B3	horizontal	27	7	52	105	74	163	21.5
	vertical	34	13	72	128	74	225	19.3
B5	horizontal	25	7	50	103	75	147	21.7
	vertical	31	2	86	126	62	235	19.4

**Table 30 materials-16-00088-t030:** Elemental analysis.

	Content (%)																
Specimen	Na	Mg	Al	Si	P	S	Cl	K	Ca	Ti	Cr	Mn	Fe	Co	Ni	Cu	Zn
B1-H	0.10	0.04	0.21	0.91	0.09	0.22	0.23	0.05	0.07	0.04	0.23	0.44	96.66	0.17	0.25	0.12	0.19
B3-H	0.10	0.05	0.21	0.91	0.07	0.36	0.21	0.05	0.06	0.03	0.12	0.38	96.71	0.18	0.16	0.08	0.32
B3-V	0.18	0.02	0.08	0.31	0.04	0.40	0.31	0.02	0.02	-	0.20	0.46	97.49	0.15	0.22	0.09	0.03
B5-H	0.02	0.07	0.32	1.36	0.09	0.33	0.07	0.08	0.09	0.04	0.15	0.44	96.09	0.13	0.21	0.07	0.47
B5-V	0.09	0.02	0.04	0.33	0.05	0.46	0.16	0.03	0.02	-	0.29	0.51	97.49	0.15	0.14	0.16	0.06

**Table 31 materials-16-00088-t031:** X-ray diffraction analysis.

Specimen	Chemical Compound	Mineral	Chemical Formula	Content (%)	PAI_α_	PAI_β_
B1-H	hydroxide—ferric oxide	lepidocrocite	γ-FeO(OH)	80	0.18	0.04
	hydroxide—ferric oxide	goethite	α-FeO(OH)	18		
	silicone dioxide	quartz	SiO_2_	1		
	hydroxide—ferric oxide	akaganeite	β-FeO(OH)	1		
B3-H	hydroxide—ferric oxide	lepidocrocite	γ-FeO(OH)	80	0.22	0.03
	hydroxide—ferric oxide	goethite	α-FeO(OH)	20		
	silicone dioxide	quartz	SiO_2_	minor		
	hydroxide—ferric oxide	akaganeite	β-FeO(OH)	minor		
B3-V	hydroxide—ferric oxide	lepidocrocite	γ-FeO(OH)	68	0.32	0.27
	hydroxide—ferric oxide	goethite	α-FeO(OH)	30		
	hydroxide—ferric oxide	akaganeite	β-FeO(OH)	2		
B5-H	hydroxide—ferric oxide	lepidocrocite	γ-FeO(OH)	83	0.19	0.01
	hydroxide—ferric oxide	goethite	α-FeO(OH)	17		
	silicone dioxide	quartz	SiO_2_	minor		
	hydroxide—ferric oxide	akaganeite	β-FeO(OH)	minor		
B5-V	hydroxide—ferric oxide	lepidocrocite	γ-FeO(OH)	65	0.46	0.03
	hydroxide—ferric oxide	goethite	α-FeO(OH)	35		
	hydroxide—ferric oxide	akaganeite	β-FeO(OH)	minor		

## Data Availability

Not applicable.

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
