# Peer review of "Experimental Measurement of Deposition Chloride Ions in the Vicinity of Road Cut"

_materials, 2022, doi:10.3390/ma16010088_

Round 1
Reviewer 1 Report
The authors present the resuls of the experimental measurements of deposition ions in the vicinity of road I/11 near the HrabynÄ› – Josefovice from 2021. For reinforced concrete structures, chloride ions cause faster carbonation of concrete, which reduces the service life of reinforced concrete structures. Observation of the microclimate in the vicinity of the roads gives to engineers a basis for the correct design of structures around the roads.The results of the measurements can help corrosion engineers and designers of structures in the vicinity of roadways to properly design structures for their service life design. The observation of chloride deposition is not only used for the control of the corrosion aggressiveness of the environment, but also for the design of steel and concrete structures. The conclusions of the experimental measurements are intended to help engineers to design a structure that is safe, serviceable and sufficiently resistant to chloride ions within its service life.
Only the following formal errors are in this document:
1) The order of the cited references is not maintained - the reference 8, 17-20
2) The reference [14] is not cited
3) IN the chap. 4.3 is following text : " [Error! Reference source not found]"
Author Response
Dear reviewer,
the authors thank you for your helpful comments on the article. References to individual publications are listed in ascending order, and an incorrect reference has been corrected.
Responses to individual comments:
1) The order of the cited references is not maintained - the reference 8, 17-20
The authors thank the reviewer for the comment. References have been edited.
2) The reference [14] is not cited
The authors thank the reviewer for the comment. The reference is used. This comment is related to comment 3), for which the authors also thank the reviewer.
3) IN the chap. 4.3 is following text : " [Error! Reference source not found]"
Reviewer 2 Report
De-icing salts or brine are widely used for safe winter traffic on the roads. However the dispersed chloride ions are nightmare for the safety operation of steel and reinforced concrete construction in the vicinity of the road. In this aspect, the work of this manuscript is important aiming at the control of the corrosion aggressiveness of the environment, and the design of steel and concrete structures. This paper presents the results of chloride deposition measurements and analyses on corrosion coupons after one year of exposure in the vicinity of a road. The results are of help for corrosion engineers and designers of structures to properly design structures for their service life in the vicinity of roads. However, the following minor issues are suggested to be considered:
Minor suggestions,
1.‘stolen’in Table 2 is better be substituted with ‘missing’.
2. The conclusions are two long and need to be abbreviated. The current conclusions section is more like a detailed abstract. Abbreviated important findings can be included in conclusion.
Author Response
Dear Reviewer,
the authors thank you for your helpful comments on the article. Both comments on the article have been incorporated.
Responses to individual comments:
1. stolen’in Table 2 is better be substituted with ‘missing’.
The authors thank the reviewer for the comment. The designation of missing measurements has been corrected.
2. The conclusions are two long and need to be abbreviated. The current conclusions section is more like a detailed abstract. Abbreviated important findings can be included in conclusion.
The authors thank the reviewer for his comments. The conclusion was revised, partially shortened and supplemented with important findings and future research objectives.
Reviewer 3 Report
Please find the attached file.

Author Response
Dear Reviewer,
the authors thank you for your helpful comments on the article. All comments on the article are incorporated.
Responses to individual comments:
-Redaction and English grammar need to be revised.
The authors thank the reviewer for the comment. English grammar has been revised.
-Improve the Abstract to give less context and more on the knowledge gap, research questions, methods, and key findings. In the abstract, please add an indication of the achievements from your study that are relevant to the journal's scope. Please be concise.
The authors thank the reviewer for the comment. The main conclusions and the main aim of the paper were added to the abstract.
-In the Introduction section, please outline the main aim, objectives, and research questions clearly and articulate the research questions to the significance of the Measurement of Deposition Chloride Ions. The review of the literature needs more updating with works to have a clear and concise state-of-the-art analysis. This should more clearly show the knowledge gaps identified and link them to the paper's goals.
The authors thank the reviewer for his comments. The introduction has been refined and the main objectives of the paper have been presented in relation to questions from other scientists concerned with the effects of NaCl on structures and nature.
-Discussions should be expanded and deepened with comparisons of other studies already published. The authors have discussed the previous scholars' work in the Introduction but this is not sufficient to support the research outcomes presented in the Results section.
The authors thank the reviewer for his comments. The discussion was expanded in its introduction. Data from other sites are discussed and referenced.
-The scale bar in figures 8,12,13,18,19 are not clear and should be corrected.
The authors thank the reviewer for his comments. Scale bar have been corrected.
-In conclusion, the main contributions of the study to the advancement of science should be highlighted. Relate your conclusions to your research questions. A limitations section should be added and the pros and cons of using the Statistic correlation analysis should be clearly outlined. Additionally, future works and impacts should be discussed.
The authors thank the reviewer for his comments. The conclusion was revised, partially shortened and supplemented with important findings and future research objectives.
-Check the references section carefully. a few of them are very old, please substitute them with newly published papers.
The authors thank the reviewer for his comments. Old references have been substituted by newest one.
Reviewer 4 Report
This paper shows an investigation on the experimental measurement of deposition chloride Ions. Overall, this paper is well organized, and this paper can be considered after the following comments being addressed.
(1) The introduction section is too week at the current form. The author should give more introduction on the investigation of chloride ingress worldwide. Besides, the author should highlight the shortage of the previous investigation and the innovation of this study.
(2) The author should give a detailed introduction on the mass transport, and the following reference may be helpful for this paper. “Characterization of sustainable mortar containing high-quality recycled manufactured sand crushed from recycled coarse aggregate” (https://doi.org/10.1016/j.cemconcomp.2022.104629)
(3) Tables 3-5 should be reformed at the form of Table rather than an image which is not clear at the current form. The related problems should be checked and corrected throughout the whole manuscript.
(4) The variance or confidence interval should be marked in Figs. 4-6.
(5) The error bar should be added in Fig. 14.
Author Response
Dear Reviewer,
the authors thank you for your helpful comments on the article. Comments that could be incorporated are incorporated. Unfortunately, comment (5) could not be incorporated, please see below for comments.
Responses to individual comments:
(1) The introduction section is too week at the current form. The author should give more introduction on the investigation of chloride ingress worldwide. Besides, the author should highlight the shortage of the previous investigation and the innovation of this study.
The authors thank the reviewer for his comments. The introduction has been refined and the main objectives of the paper have been presented in relation to questions from other scientists concerned with the effects of NaCl on structures and nature.
(2) The author should give a detailed introduction on the mass transport, and the following reference may be helpful for this paper. “Characterization of sustainable mortar containing high-quality recycled manufactured sand crushed from recycled coarse aggregate” (https://doi.org/10.1016/j.cemconcomp.2022.104629)
The authors thank the reviewer for his comments. Information on the effect of chlorides on concrete structures has been added to the introduction of the paper.
(3) Tables 3-5 should be reformed at the form of Table rather than an image which is not clear at the current form. The related problems should be checked and corrected throughout the whole manuscript.
The authors thank the reviewer for his comments. The tables have been modified.
(4) The variance or confidence interval should be marked in Figs. 4-6.
The authors thank the reviewer for his comments. Information about the 95% confidence interval (grey area) has been added to the graph description.
(5) The error bar should be added in Fig. 14.
The authors thank the reviewer for his comments. Error bar cannot be added because there is just one corrosion coupon for each value. Thus, it is not possible to determine the variance of the values. It is the value of one particular measurement.
Round 2
Reviewer 3 Report
All the figures which contain more than one image should be labeled as a) b) c)...
for example, there is a mistake in the scale bar in Fig 19. and it isn't easy to address it, so I have to explain in this way:
Fig.19 200X magnification right side image (which could be simply called 19(d) ) the scale bar is 100 μm which you put 1000 μm, please fix it and also change the scale from μm to mm, 1 mm is easier to read rather than 1000 μm, please apply this change in all the related figures accordingly.
The rest of the comments have been done correctly.
Author Response
The authors thank the reviewer for his helpful comments. The authors have incorporated the comments.
Responses to specific comments:
All the figures which contain more than one image should be labeled as a) b) c)...
- The authors thank the reviewer for the comment. Figures with multiple images are labelled according to the comment.
for example, there is a mistake in the scale bar in Fig 19. and it isn't easy to address it, so I have to explain in this way:
Fig.19 200X magnification right side image (which could be simply called 19(d) ) the scale bar is 100 μm which you put 1000 μm, please fix it and also change the scale from μm to mm, 1 mm is easier to read rather than 1000 μm, please apply this change in all the related figures accordingly.
- The authors thank the reviewer for the comment. The scale error has been corrected. At the same time, the units were changed from μm to mm as recommended.